# Immunological Interactions between Intestinal Helminth Infections and Tuberculosis

**DOI:** 10.3390/diagnostics12112676

**Published:** 2022-11-04

**Authors:** Khethiwe Nomcebo Bhengu, Pragalathan Naidoo, Ravesh Singh, Miranda N. Mpaka-Mbatha, Nomzamo Nembe, Zamathombeni Duma, Roxanne Pillay, Zilungile L. Mkhize-Kwitshana

**Affiliations:** 1Department of Medical Microbiology, School of Laboratory Medicine and Medical Sciences, College of Health Sciences, Nelson R. Mandela School of Medicine, University of KwaZulu-Natal, Durban 4001, South Africa; 2Division of Research Capacity Development, South African Medical Research Council (SAMRC), Cape Town 7505, South Africa; 3Department of Biomedical Sciences, Faculty of Natural Sciences, Mangosuthu University of Technology, Umlazi, Durban 4031, South Africa; 4Department of Medical Microbiology, School of Laboratory Medicine and Medical Sciences, College of Health Sciences, Howard College, University of KwaZulu-Natal, Durban 4041, South Africa

**Keywords:** *Mycobacterium tuberculosis*, helminths, coinfection, immune response, Bacille Calmette-Guerin, vaccination

## Abstract

Helminth infections are among the neglected tropical diseases affecting billions of people globally, predominantly in developing countries. Helminths’ effects are augmented by coincident tuberculosis disease, which infects a third of the world’s population. The role of helminth infections on the pathogenesis and pathology of active tuberculosis (T.B.) remains controversial. Parasite-induced suppression of the efficacy of Bacille Calmette-Guerin (BCG) has been widely reported in helminth-endemic areas worldwide. T.B. immune response is predominantly proinflammatory T-helper type 1 (Th1)-dependent. On the other hand, helminth infections induce an opposing anti-inflammatory Th2 and Th3 immune-regulatory response. This review summarizes the literature focusing on host immune response profiles during single-helminth, T.B. and dual infections. It also aims to necessitate investigations into the complexity of immunity in helminth/T.B. coinfected patients since the research data are limited and contradictory. Helminths overlap geographically with T.B., particularly in Sub-Saharan Africa. Each disease elicits a response which may skew the immune responses. However, these effects are helminth species-dependent, where some parasites have no impact on the immune responses to concurrent T.B. The implications for the complex immunological interactions that occur during coinfection are highlighted to inform government treatment policies and encourage the development of high-efficacy T.B. vaccines in areas where helminths are prevalent.

## 1. Introduction

Intestinal helminths are parasitic worms infecting over 1.5 billion people globally [1] Most helminth cases occur in tropical and sub-tropical areas such as Sub-Saharan Africa, the Americas, China and East Asia [1]. Humans are infected with helminth parasites after ingesting eggs or larvae from contaminated water, soil or food or through active skin penetration by infective hookworm larvae in contaminated soil [2]. Climate change, malnutrition, overcrowding, poverty and poor sanitary conditions are risk factors associated with the high helminth prevalence in Africa and other developing countries, making effective treatment and the eradication of infection challenging [1,2,3,4]. The most common intestinal helminth species infecting humans are *Schistosoma mansoni, Trichuris trichuria* (whipworm), *Ascaris lumbricoides* (roundworm), *Necator americanus* and *Ancylostoma duodenale (hookworms)* [1,2].

Tuberculosis (T.B.) is an infectious bacterial disease caused by different strains of acid-fast bacilli belonging to the *Mycobacterium tuberculosis (Mtb)* complex [5]. The T.B. bacteria are airborne, and transmission occurs when a T.B.-infected person coughs, sneezes or spits, expelling the infected droplets into the air. Inhalation of these aerosols may result in infection of the next host [6]. T.B. continues to be a public health problem across the world, with the World Health Organization (WHO) reporting over 10 million T.B. cases in 2020 [7]. Approximately 1.5 million TB-related deaths were reported worldwide in 2020 [7]. Globally, Africa accounts for 50% of cases of T.B. and human immunodeficiency virus (HIV) coinfection [7]. Furthermore, in Africa, T.B. is commonly observed in HIV-infected patients, and it is the leading cause of death among them [7].

T.B. exposure results in the initiation of an immune response to fight the infection. The immune response to T.B. involves the interaction of innate and adaptive immune responses. It is dependent on the cellular immune response, which is mediated by proinflammatory T-helper type 1 (Th1) and Th17 cells [8,9,10]. The Th1 cytokines, which are interferon-γ (IFN-γ), interleukin 12 (IL12) and tumor necrosis factor-α (TNF-α) and Th17 cytokines (IL-17, IL-21, IL-22 and IL-23) play a role in combating bacterial and viral infections [8,9,10]. Helminth exposure, on the other hand, induces an anti-inflammatory Th2 immune response which is characterized by the production of cytokines such as IL-4, IL-5, IL-9, IL-10 and IL-13, and increased levels of circulating immunoglobulin E (IgE) antibodies, eosinophils, and mast cells, regulatory T cells (Tregs) and transforming growth factor-β (TGF-β) [11,12].

T.B. commonly overlaps geographically with soil-transmitted helminths, especially in developing countries [13,14,15,16], and this co-endemicity has implications for public health and the afflicted hosts. Helminth infection-induced immune responses could promote the pathogenesis of severe T.B. infections [16,17,18]; others report that they can also be beneficial in reducing T.B. severity [19,20,21,22]. However, there is no conclusive evidence to confirm whether helminth-induced immunity modulates T.B.-specific immune responses or vice-versa, and studies have yielded contradictory results. Therefore, knowledge on the interaction between T.B. and helminth infections is limited, as are the available data.

Given the current evidence on potential immunologic implications, such as those that could influence T.B. vaccination, treatment and diagnosis, more research is needed to determine the influence of helminth coinfection on T.B. control and how to negate any adverse effects. As a result, this review will summarize what is currently known about T.B. and helminths’ immune responses in human and experimental studies, both separately and in the context of coinfection. The review will also elucidate the effects of T.B. and helminth coinfections on vaccine efficacy and the implications for long-term health care.

## 2. Article Search Strategy for the Current Review

An electronic search of online databases such as Google Scholar, Google, PubMed, Science Direct, online library sources, and Web of Science were utilized to extract research and review articles using phrases and words: helminth, tuberculosis, helminth and tuberculosis coinfection, helminth and tuberculosis vaccine and helminth and tuberculosis diagnosis in humans, animals and in vitro studies. A PRISMA flow diagram of the search strategy and research design process for this review is presented in Figure 1.

## 3. The Host Immune Response to Helminths

Helminths are parasitic and multicellular organisms that coevolved with their hosts [23]. These parasitic infections are often asymptomatic, but there are cases of heavy worm burden. These have been linked to persistent health conditions such as anemia, fatigue, growth stunting and poor cognitive development [24]. Helminths are the driving force behind how immunity is initiated and maintained [25]. They typically create long-term infections in their hosts. They have the power to influence physiological and immunological homeostasis to ensure their continuing existence [25].

Helminths mature within the infected subject and lay eggs for transfer to another host, exposing them to multiple stages of parasite development, each of which elicits a unique immune response [26]. Helminths have evolved to exploit a range of host immunoregulatory mechanisms and activate generic suppressive pathways that can suppress bystander responses to other antigens, allergens, and self-antigens [12]. Helminths have been dubbed “masters of immunoregulation” because of their capability to control immunity to escape being eliminated by the host [25,27]. Helminths enter the body through the skin or intestinal epithelium’s barrier surface, where they block the transcription of numerous molecules that keep the epithelium intact [28].

Tissue injury activates the production of “alarmins” (IL-33 and thymic stromal lymphopoietin (TSLP)) and the identification of invaders by pattern recognition receptors (PPRs) in the host [28]. The Th1 proinflammatory cytokine production is driven by pattern recognition receptors (PRRs) such as toll-like receptors (TLRs) or C-type lectin receptors (CLRs), whereas IL-33 and TSLP initiate a Th2 anti-inflammatory response [28].

Helminths stimulate increased mucin synthesis, smooth muscle contractility and epithelial cell turnover as a host defense to eliminate the infection. There is also increased IgE and IgG1 production in mice and IgE and IgG4 production in humans [12,28]. All these processes work together to drive worm expulsion and wound-healing responses, which control worm-induced tissue damage [28].

The Th2 immune response induced by helminths includes interleukins (IL-4, IL-5, IL-9, IL-10, and IL-13), broad or localized eosinophilia and hyperplasia of goblet and mucosal mast cells [12,28]. The CD4-positive Th2 cells were initially identified as an essential source of IL-4, IL-5, IL-9, IL-10 and IL-13 cytokines [29]. Eosinophils, basophils and innate lymphoid cells (ILCs) can also produce some of these cytokines in response to helminth infections [29]. Although the Th2 immune response induced by helminth parasites is stereotypical, the initiation, progression and culmination of this response require interaction with different cell types, most notably: epithelial or stromal cells, ILCs, antigen-presenting cells, dendritic cells, macrophages, T cells, B cells, eosinophils, mast cells and basophils [12].

Tregs maintain the Th2 dominance, IL-10 and TGF-β, which mediate the suppression of competing Th1 and Th17 cell populations [30]. Tregs modulate the immune system to prevent tissue damage induced by proinflammatory responses, maintain tolerance to self-antigens and abrogate autoimmune disease [31]. These cells can be divided into two subsets: natural Tregs that develop in the thymus, and induced Tregs that arise from conventional CD4 positive T cells in the periphery, which are promoted by chronic antigen exposure [32]. The forkhead/winged-helix transcription factor (Foxp3) is a crucial marker for identifying these subsets, but it may be expressed on activated CD4 positive T cells [32].

Helminth-induced suppression of immunopathology also involves CD4+ Tregs (Foxp3+ or Foxp3), CD8+ Tregs, regulatory B cells (Bregs), IL-4-responsive cells, TGF-β, and IL-10 [33]. Since an increased Th2 response can potentially induce disease, a regulated response must be generated. This is referred to as the modified Th2 cell response and is characterized by the downregulation of Th2 cytokines [12].

According to the hygiene hypothesis, in developed countries where sanitation is good, and helminths have been eliminated, there is an increase in allergic diseases such as asthma and allergic rhinitis, and autoimmune diseases such as Crohn’s disease [27]. This hypothesis has led to many human and animal studies conducted using live helminth parasites to determine whether helminths do nullify the effect of allergies and autoimmune disorders. Human studies conducted in underdeveloped countries where helminths are still prevalent showed fewer allergies and autoimmune diseases [27,34,35]. Others have reported evidence of decreased allergies in developing countries [36].

Helminths induce various immune and physiologic modifications to survive the hostile immune response directed against them and their general survival. These survival mechanisms include this modified Th2 response [27]. These parasites also promote angiogenesis, which changes tissue vascularity and thus provides a good niche for their survival [37]. The overall immune modulation of helminths invokes immunosuppression, immunologic and physiological tolerance and a modified Th2 response [27]. These can lead to a reduced immune response, thus amplifying susceptibility to infection with other pathogens, reduced anti-tumor immunity and reduced vaccine efficacy. The host immune response profile to helminth infection is presented in Figure 2.

Figure 2 Footnotes: IL: interleukin; IFN-γ: interferon-gamma; TGFβ: transforming growth factor beta; TNFα: tumor necrosis factor-alpha; ILCs: innate lymphoid cells; TSLP: Thymic stromal lymphopoietin; AAMs: alternatively activated macrophages; DAMPS: damage-associated molecular patterns; PAMPS: pathogen-associated molecular patterns. Red arrow pointing up indicates cytokines that are upregulated/increased during the early stages of helminth infection and those that are upregulated during the chronic stages.

## 4. The Host Immune Response to T.B.

T.B. enters the body via inhaled droplets to the alveoli. It interacts with the alveolar macrophages, infecting and multiplying inside them, thus making these cells the first line of defense against infection [6]. In immunocompetent individuals, macrophages are activated, and they phagocytose and remove T.B.

In some cases, the disease is controlled and kept in an inactive or latent state in distinct foci known as granulomas bacteria [9,15,38,39]. However, some bacteria can escape this fate, multiply and eventually cause an active infection. This may be due to the intrinsic capacity of the macrophage, the immune status of the host or the virulence of the infecting bacteria [9,15,38,39]. *Mtb* is, therefore, a pathogen that can cause both latent and active disease [40].

### 4.1. Innate Responses to T.B.

The initial stages of T.B. infection include phagocytosis of the bacteria by macrophages [6]. Receptors that recognize a broad spectrum of mycobacterial ligands cause phagocytosis [9]. Pathogen recognition receptors, TLRs, complement receptors (C.R.), Nucleotide Oligomerization Domain (NOD)-like receptors and C-type lectins have all been implicated in recognition of mycobacteria and the initiation of the cytokine response [8].

When phagocytic cells encounter T.B., they get activated and generate cytokines, including proinflammatory cytokines such as TNF-, IL-1, IL-6, IL-12 and IFN-γ [8]. Increased susceptibility to T.B. was reported to be linked to genetic abnormalities in IFN-γ production [15,41]. IFN-γ is involved in activating macrophages that fight mycobacteria through intracellular killing and antigen presentation to T lymphocytes [42]. Vitamin D is also involved in killing *Mtb*, which is aided by the creation of peptide cathelicidin [43].

The presentation of T.B. antigens by dendritic cells in lymph nodes, possibly aided by neutrophils, initiates a local immune response that culminates in pathogen killing by reactive oxygen species (ROS) and antimicrobial peptides [8].

Cells required in the host’s defense against *Mtb* include monocytes, macrophages, neutrophils, natural killer (NK) cells and dendritic cells. Together, these cells form a primary granuloma, which may allow *Mtb* growth while containing the infection until T cells are recruited to the infection site, a response process that takes weeks [8]. Phagolysosomal fusion, reactive oxygen and nitrogen intermediates, and antimicrobial peptides such as cathelicidin induced by vitamin D are innate mechanisms against *Mtb* [43].

NK cells may eliminate intracellular *Mtb* through the activation of perforin, where the antimycobacterial factor granulysin binds to the bacterial cell surface and disrupts the membrane, resulting in bacterial osmotic lysis [44]. Apoptosis is a critical mechanism for the infected host cell to limit *Mtb* replication to a minimum. Phagocytic cell apoptosis may prevent the spread of disease, diminish the viability of intracellular mycobacteria and reduce the risk of infection [45].

### 4.2. Adaptive Immune Responses to T.B.

Adaptive immunity develops after exposure to mycobacterial antigens or vaccination with BCG. This part of the immune system is triggered when the innate immune response is insufficient to suppress T.B. infection. The control of T.B. requires Th1 immune responses (IFN-γ, IL-12 and TNF-α) and Th17 responses (IL-17 and IL-23). Th1 responses are proinflammatory and develop a cell-mediated reaction [38]. Th1 cells produce IFN-γ through the T-box transcription factor (TBX21). Both IL-12 and IFN-γ are the leading cytokines in Th1 responses, where IL-12 is secreted by antigen-presenting cells [39,46]. The IL-12 receptor, which is expressed on the surface of T cells, interacts with IL-12. The increased T-bet (encoded by TBX21) boosts the signal transducer and activator of transcription 4 (STAT4), a regulator of Th1 cells [46].

T-bet binds to and affects the expression of Th1-specific genes and Th1 and Th17 cell expression [46]. This is important since the control of T.B. requires Th1 responses. STAT4 and T-bet work together to ensure optimal IFN-γ levels, and their depletion eliminates IFN-γ production [46].

T.B. immunity involves many cells, such as T cells, B cells and natural killer (NK) cells, with CD4+ T cells being the primary cell type in T.B. control [47]. The CD4+ Th1 cells are central to the control of T.B.; these cells secrete IFN-γ and TNF-α, which are both critical in the management of T.B. [38]. IL-12 regulates the induction of IFN-γ, and mutations in the genes coding for IL-12, IL-12R, IFN-γR or STAT1 or depletion of CD4+ T cells (as seen in HIV infection) all promote susceptibility to disseminated T.B. [38]. IFN-γ stimulates phagocytosis, phagosome maturation, the production of reactive oxygen intermediates (ROS) and antigen presentation in macrophages.

IFN-γ is regarded as the primary cytokine that regulates T.B. infection and eradication. It works by activating the infected macrophage, resulting in the production of reactive oxygen and nitrogen species, which have a microbicidal role [48]. In terms of memory immune responses, CD4+ Th17 cells and Th1 cells have been identified as enhancing the host’s resistance to T.B. [49]. Th17 cells are a lineage of CD4+ T helper cells that produce the cytokine IL-17, IL-17F and IL-22, and they play a role in developing an optimal Th1 response [50].

Th17 was first described as a distinct population of the T helper cells controlled by the transcription factor RAR-related orphan receptor gamma (RORyt) [51]. They develop independently of T-bet, STAT4, GATA-3 and STAT6 transcription factors critical for the development of Th1 and Th2 development, respectively [51]. The central effector cytokines of Th17 are IL-17; other cytokines are IL-22 and IL-26 [52]. The immune response to T.B. infection is directed mainly by a Th1 response, with contributions from Th17 and other cells. A strong proinflammatory milieu also characterizes T.B. infection.

On the other hand, human innate immune responses to *Mtb* infection are still poorly understood, owing to the limitations in examining pulmonary-specific immunity. 

Therefore, understanding the interaction of innate and adaptive immune cells in human T.B. is crucial for identifying new immunomodulatory targets and clarifying protective immunity processes. The immune response profiles to tuberculosis infection are presented in Figure 3.

Innate and Th1-dominant adaptive immune responses interact to produce granulomas. Innate and adaptive immune responses are critical for microorganism eradication.

Figure 3 footnotes: IL: interleukin; IFN-γ: interferon-gamma; TNF-α: tumor necrosis factor-α; ROI: reactive oxygen intermediates; RNI: reactive nitrogen intermediates. Red arrow pointing up indicates cytokines that are upregulated/increased during T.B. infection. Red arrow pointing down indicates cytokines that are downregulated during T.B. infection.

## 5. Host Immune Response during Helminth Coinfection with T.B.

The geographic distributions of helminths and T.B. overlap substantially, particularly in underdeveloped countries, resulting in an increased likelihood of coinfection with both pathogens [15,16]. This coexistence has also led to the hypothesis that helminths can worsen the effects of T.B. There have been suggestions that the anti-inflammatory response induced by helminths in cases of coinfection might dampen protective and immunopathological responses to T.B. [15,16].

An Ethiopian study investigated the association between intestinal helminths and active T.B. and found that helminth infection increases the likelihood of developing active T.B. [53]. This and other studies also suggested that patients with coinfection may have antagonistic effector cell responses in responding to and regulating these diseases [30,54]. This can also imply that the efficacy of the vaccines may be reduced.

One school of thought suggests that helminths create an environment that weakens the host’s defenses against T.B. By activating the IL-4 receptor pathway, a preexisting helminth infection inhibits an innate pulmonary anti-T.B. defense [55]. In coinfected mice models, helminth-induced lung alterations increased susceptibility to T.B. [55]. Macrophages can be classically or alternatively activated. Classically activated macrophages (CAMs) increase the activity of nitric oxide synthase (iNOS), which converts L-arginine to nitric oxide and citrulline. Nitric oxide promotes intracellular *Mtb* killing.

On the other hand, alternatively activated macrophages (AAMs) induce arginase, which competes with iNOS for L-arginine, thereby reducing nitric oxide production for the intracellular killing of *Mtb* [48]. *Mtb* resistance in helminth-infected mice is promoted by AAMs. This major cellular pathway compromises the helminth-infected host’s ability to limit *Mtb* growth [55].

A review in support of this proposed role of the Th2-dominant phenotype on *Mtb* control illustrated that AAMs might inhibit the macrophage killing of *Mtb* [48]. Conversely, a murine study in South Africa using *Nippostrongylus brasiliensis* (*Nb*) revealed that *Mtb* colonies were reduced in the lungs of *Nb*-infected mice. The stimulation of pulmonary CD4+ T cells and Th1 and Th2 cytokines, neutrophils and alveolar macrophages was elevated. This suggests that *Nb* infection triggers a macrophage response, which protects the host throughout the early phases of mycobacterial disease and subsequent illness [19].

Both helminths and T.B. have independent mechanisms for initiating the host immune response, with significant consequences for the immunology of each infection [15,16]. The coexistence of helminth infection and active tuberculosis has been demonstrated in epidemiological, cross-sectional and case-control studies that looked at the prevalence and correlation of the two diseases. Pulmonary T.B. patients were found to have a significant rate of intestinal nematode infection, indicating that helminth immunomodulation may affect the control of T.B. [53,56].

In Ethiopia, some studies reported an increase in the prevalence of helminth coinfection in T.B. patients, where one study found a higher risk of parasites among active T.B. patients than in healthy community controls [17,57,58]. Likewise, in Iran, a higher prevalence of intestinal helminths was found in tuberculosis patients compared to the uninfected subjects [59]. Taghipour and colleagues also determined that immunocompromised T.B. patients are more vulnerable to parasitic gastrointestinal infections [60]. It was reported that Blastocystis subtype 1 was the most common subtype found in T.B. patients; however, a phylogenetic analysis revealed no distinction between Blastocystis isolates from T.B. patients and those from the uninfected [59].

*S. mansoni* was also a risk factor for T.B. infection, and it altered the clinical presentation and pathogenesis of T.B. in Tanzania [61]. The authors recommended treatment of this parasite using praziquantel in T.B. infection management [61].

A systematic review suggested that health education be implemented to help prevent intestinal helminth infection. It further added that screening for helminths should be possibly included in the treatment strategies for tuberculosis patients [59]. Another review suggested an association between *Toxoplasma gondii (T. gondii*) seropositivity and having tuberculosis, with *T. gondii* seropositivity, which indicates chronic infection, being relatively common among tuberculosis patients [62].

*Strongyloides stercoralis* coinfection with pulmonary T.B. was implicated in the cause of the skewed immune response to mycobacterial disease [63]. The proinflammatory Th1 cytokines were reduced, whereas the anti-inflammatory Th2 and Th3 cytokines were elevated, thus leading to a conclusion that helminth coinfection may modulate protective immune responses in latent T.B. [63]. A study of immunological correlates in T.B. coinfection with *S. mansoni* in Kenya, on the other hand, discovered that the expression of T.B.-specific Th1 cytokines was maintained. Individuals with latent tuberculosis and *S. mansoni* infection had more CD4+ Th1 cells than those who were only latently T.B.-infected [22]. There were similar results in a Brazilian study, whose findings revealed that A. *lumbricoides* infection had no impact on Th1, Th2 and Th17 responses or the T cell populations [21].

A Th1 immune response observed during persistent filarial infection was characterized by a reduction in Purified Protein Derivative (PPD)-specific IFN-γ and IL17 responses [64]. The study suggested that filaria infection reduced the PPD-specific IFNγ and IL17 responses. In addition, it was observed that onchocerciasis patients’ peripheral T cells had a weak response to *Mtb* antigens [65]. Elias and colleagues illustrated that compared to dewormed patients, helminth-infected individuals displayed low Th1 immune response and IFN-γ production in response to mycobacteria infection [66]. Lastly, it has been suggested that a robust Th1 response characterizes cell mediated protection against T.B. infection, and coinfection with helminths could modulate these immune responses by driving Th2 and Treg cells [17,67].

Furthermore, enhanced Treg function is associated with helminth infection and may suppress Th1 responses against unrelated antigens [12,67]. This finding was supported by studies which showed that intestinal helminth coinfection was associated with a reduced Th1 response in active T.B. [16,68]. Type I immunity and their proinflammatory cytokines such as IFN-γ, IL-12 and TNF-α have a protective role against *Mtb*. By contrast, the induction of type 2 immunity, e.g., Th2 and Treg cells (as seen in helminth infections) and their anti-inflammatory cytokines, were reported to suppress the efficient immune response against T.B. [38].

A mouse model study of Schistosoma mansoni showed a reduced protective efficacy of BCG vaccination against *Mtb* [66]. Another study demonstrated that concomitant helminth infections significantly impair the immunogenicity of BCG vaccines, an impairment associated with increased TGF-β production [30]. During active T.B., asymptomatic helminth infection has been shown to have a considerable impact on host immunity in a double-blind, randomized clinical study [17]. In comparison to the placebo group, eosinophils and IL-10 levels decreased after albendazole treatment [17]. Another albendazole treatment study was conducted to determine the immunological effects of deworming on proinflammatory cytokine responses to plasmodial antigens. The study demonstrated improvements in immune hypo responsiveness, where anthelmintic treatment significantly increased proinflammatory cytokine responses to Plasmodium falciparum-infected red blood cells [69].

In Egypt, it was determined that hookworm infection was one of the risk factors for the failure of T.B. therapy [70]. However, a human study in the United Kingdom (U.K.), where the authors studied migrants from Nepal, found that hookworm infection reduced T.B. growth and may reduce the risk of infection [20]. According to the evidence presented above, some studies demonstrated that helminthiasis has a negative impact on T.B. diseases, while others showed a beneficial effect. Table 1 summarizes some of the studies investigating helminth and T.B. coinfections.

Although HIV is not covered in this review, there is evidence of a concurrent distribution of triple disease burden involving tuberculosis, helminths and HIV, particularly in Sub-Saharan Africa. This necessitates a greater focus on disease management strategies by various policymakers [71].

## 6. Effect of Helminth Infection on T.B. Vaccine

BCG is currently the only T.B. vaccine available; it celebrated its 100th anniversary in 2021. Alternative vaccines are being developed [83]. The BCG vaccine is still the only option for protection against human T.B., and it is inexpensive, safe and widely available. BCG effectiveness against T.B., however, varies in the high helminth-burden areas of the world [83]. Children are typically given the BCG vaccine. A review reported that BCG could provide protection against severe forms of T.B., including meningitis and miliary [84].

The BCG vaccine is administered to more than 80% of all newborns and babies in countries where it is included in the national childhood immunization program; however, it does not prevent the development of latent tuberculosis or the reactivation of pulmonary disease in adults [85]. BCG has been reported to be less effective in T.B.-coinfected individuals living in helminth-endemic areas [64]. However, another study reported no difference in BCG vaccination status and tuberculin skin testing (TST) responses in patients with or without T.B. and helminth coinfection [67].

An Ethiopian study found that helminth infection influenced BCG vaccination outcomes, and PPD-specific cellular immune responses improved in helminth-treated individuals compared to untreated controls [64]. Deworming was shown to boost the efficacy of BCG immunization in this randomized experiment [64]. In addition, it was found that the BCG vaccination of PPD-negative individuals in a helminth-infected population in Ethiopia had poor immunogenicity, and they concluded that this was due to a high Th2 bias in immunological responses caused by chronic helminth infection [64].

Furthermore, in another study, *S. mansoni* was found to reduce the protective efficacy of BCG vaccination against *Mtb*, possibly by attenuating protective immune responses to mycobacterial antigens and polarizing general immune responses to a Th2 profile [66]. 

Th2-like IL-10 responses elicited by intestinal helminths may interfere with Th1-like IFN responses induced by BCG, altering the protective immune response to BCG vaccination [86]. The impact of helminth infection is due to the antigen-specific modification of cell-mediated immunity, and the diminished efficacy could be owing to impaired immune responses to recall antigens [87].

Furthermore, helminth infection during pregnancy has been shown to persist into childhood and shift immunity away from Th1 responses, which are required in T.B. infection and vaccination [72]. Chronic helminth infections increase susceptibility to T.B. infections requiring Th1 responses and also lead to impaired efficacy of the BCG vaccine [30,88].

While there is mounting evidence that helminth prophylaxis could have a role in combating the HIV/AIDS and T.B. pandemics [89], observational research and randomized controlled trials have not revealed a uniform clinical picture. Deworming programs may help to enhance community-based health measures such as proper sanitation, access to clean water and adequate education [90]. More intervention research is required to demonstrate the impact of deworming on tuberculosis disease progression.

## 7. Helminth and T.B. Coinfection-Immune Mediated Pathology

The typical immune response to helminths, characterized by decreased IFN-γ, reduced T cell proliferation and IL-2 as a result of increased Th2/Treg cytokines, attenuates a potent anti-tuberculosis IFN-γ immune response and therefore uncontrolled T.B. pathology [15]. Furthermore, the helminth-induced expansion of AAMs and nitric oxide synthase suppression could also contribute to the impaired intracellular killing of T.B. in macrophages, thereby enhancing T.B. disease process [15]. In addition, the helminth-induced anergy of cognate and bystander T cells and increased apoptosis further impair T.B. responses and increase the pathogenesis [88].

## 8. Effect of Deworming during T.B.-Helminth Coinfection

The effects of deworming can be used to determine the impact of helminth infections. It was shown that the use of anthelminthic drugs to treat patients with helminths resulted in increased T cell proliferation and IFN-γ production of PBMC stimulated with PPD. The study showed that T cell responses to PPD were improved in filarial-infected patients treated with diethylcarbamazine [55,65].

The treatment of helminth-infected patients with albendazole during BCG vaccination increased proliferative and IFN-γ responses to PPD, suggesting that persistent helminth infection during BCG vaccination may contribute to a decreased T cell response to mycobacterial antigens. This meant that removing helminths via anthelminthic treatment would reduce Th2 cell and cytokine inhibitory effects on Th1 responses [91].

Toulza et al. found that anthelminthic therapy altered antimycobacterial immune responses in U.K. migrants. Patients with helminth infection had a higher frequency of CD4 + Fox P3 + T cells (Tregs) and a lower frequency of CD4 + IFN-γ + T cells, but these effects were reversed after treatment [68].

Another study in Gabon found that anti-helminth treatment with praziquantel against *Schistosoma* infection resulted in a significant decrease in CD4 + Fox P3 + T cells after treatment [92]. Since helminth infections cause widespread immunological alterations that revert to normal after the helminth infection is eradicated, their role in the interaction between their host and other pathogens could be substantial [93]. 

From the above, it is apparent that concurrent helminth and T.B. infections have demonstrated various effects on the host. These reactions could be due to different helminth species, their location in the body, different life cycles, variable (excretory/secretory) E/S products and *Mtb* infection. The virulence and infection route of the mycobacterial strain may also contribute.

Some in vitro studies have been reported to have shown that helminth infection affects *Mtb* infection in terms of immune response and disease severity, but the clinical and treatment outcome is unknown, possibly due to underpowered studies, the type or intensity of the infecting helminth and the various methodologies used to detect helminth infection [15].

## 9. Concluding Remarks

Concurrent helminth infection and T.B. both produce antagonistic immune responses. Helminths have the potential to impair the host’s ability to respond to bystander infections such as T.B. Helminth and T.B.’s spatial overlap may impair the host’s ability to respond to mycobacterial conditions. Th1 responses are required for T.B. immunity, whereas helminths mount an opposing Th2 response, which tends to dominate and thus skew the immune response. Furthermore, chronic helminth infections impair innate and adaptive immune responses to T.B. and induce immunoregulatory responses, lowering T.B. immunity even further. However, whether these opposing immune responses in helminth and T.B. coinfection affect pathological outcomes is unclear.

In helminth-endemic areas, it is suggested that chronic helminth infections reduce the efficacy of BCG, the currently available T.B. vaccine. There is conflicting evidence regarding the effectiveness of regular anti-helminth medication in the treatment of T.B., and this requires further investigation. Clarification of the effect of deworming in concurrent helminth-T.B. infections may aid in the development of government treatment policies. Since vaccines can prevent T.B. infection, the co-occurrence of helminths and T.B. must be considered when developing new vaccines and conducting research on them. Finally, more research is needed to understand better the effects of multicellular coinfecting pathogens on immune responses.

## Figures and Tables

**Figure 1 diagnostics-12-02676-f001:**
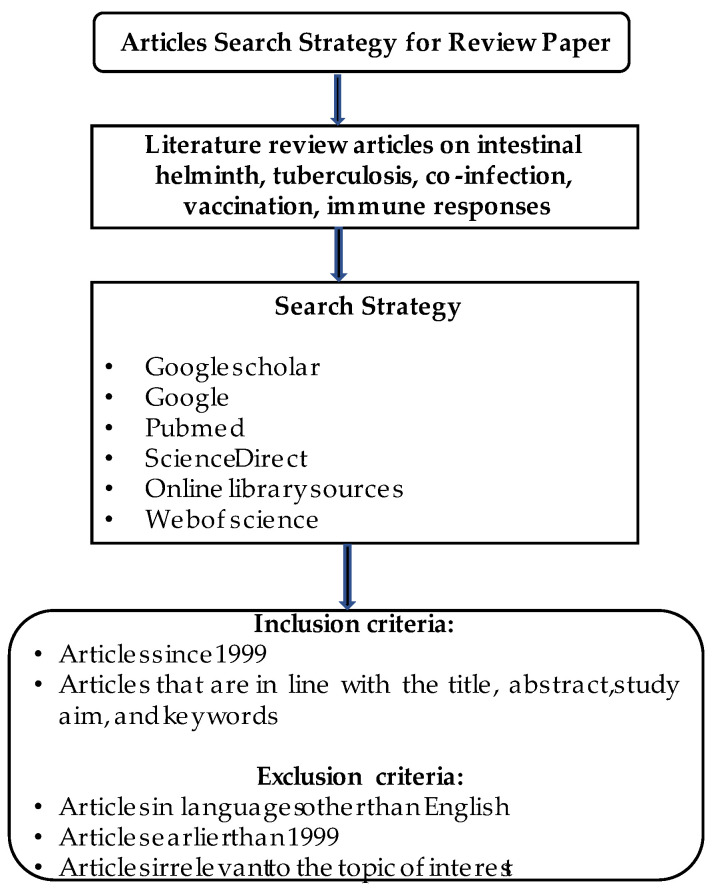
PRISMA flow diagram of the search strategy and the research design process.

**Figure 2 diagnostics-12-02676-f002:**
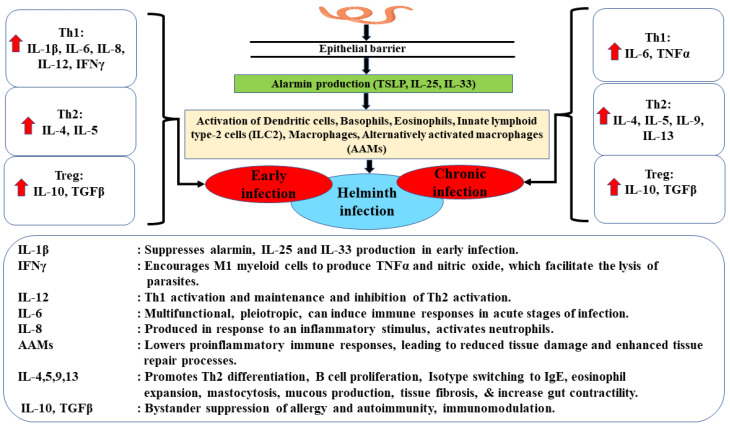
Immune response profiles during helminth infection. Migration of helminths damages epithelial barrier cells and tissues, triggering an immune response. Helminths produce damage and pathogen-associated molecular patterns (DAMPS and PAMPS). DAMPS and PAMPS activate various cells, such as epithelial, which release alarmins such as Thymic stromal lymphopoietin (TSLP), IL-25 and IL-33. Alarmins stimulate innate lymphoid cells (ILCs), aiding collagen deposition and tissue repair, and are a source of IL-5 required for eosinophil activation. Eosinophils enter tissues during helminth infection-induced inflammation. Eosinophilia is a crucial feature of the host response to helminth infection. Alarmins promote B cell activation and induction of alternatively activated macrophages (AAMs). AAMs stimulate IL-10 and TGF-β, which reduce the host’s immune response to pathogens to avoid damaging the host and maintain normal tissue homeostasis. Classically activated macrophages, stimulated by IFN-γ produce proinflammatory cytokines (IL-1β, IL-6, IL-8, IL-12 and TNF-α).

**Figure 3 diagnostics-12-02676-f003:**
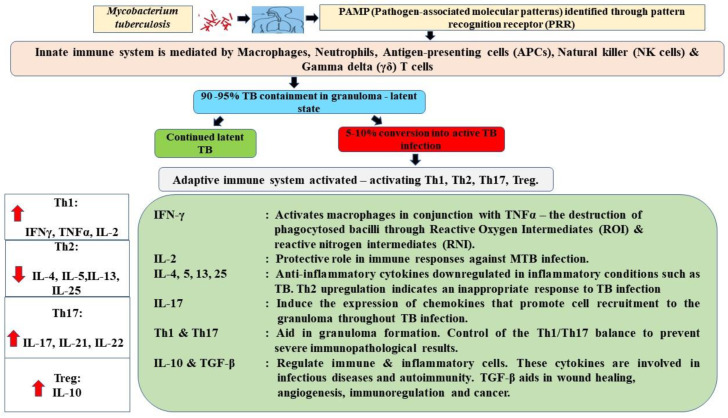
Immune response profiles during tuberculosis infection. Mycobacteria encounter alveolar macrophages where they are phagocytosed, kept inside phagosomes and exposed to antimicrobial peptides and degrading lysosomal enzymes (lysozyme). However, pathogenic mycobacteria have developed strategies to subvert the host’s defenses. Th1-cell activity (IFN-γ, IL-12 and TNF-α) is required for *Mycobacterium tuberculosis* immunity. IFN-γ activation of macrophages promotes bacterial killing by forming toxic reactive oxygen intermediates (ROI) and reactive nitrogen intermediates (RNI). An array of cytokines and chemokines, including tumor necrosis factor (TNF-α), induces a proinflammatory response and direct immune cells to the infection site. Dendritic cells migrate to draining lymph nodes, where they encounter many immature T cells. In the presence of proinflammatory cytokines such as IFN-γ and IL12, T cells become activated, multiply and differentiate into T helper (Th)1 cells. IFN-γ stimulates macrophages and triggers the potent antimicrobial activities of the primed Th1 cells.

**Table 1 diagnostics-12-02676-t001:** Summary of experimental and human studies focusing on helminth and tuberculosis coinfections.

References	Study Type, Location and Helminth(s)	Study Aim	Major Findings
[72]	Human study in Kenya. *Wuchereria bancrofti* and *Schistosoma haematobium*	To investigate whether prenatal immunity to helminths persists in childhood and if it alters the immune response to BCG	Compared to patients who had prenatal sensitization 10–14 months after BCG immunization, T cell IFN-γ production was 26-fold higher in infants who were not sensitized to filariae or schistosomes in utero.
[73]	Human study in South Africa. *Ascaris lumbricoides* and *Trichuris trichiura*	To determine total serum IgE before and after tuberculosis therapy	T.B. therapy resulted in reduced serum *Ascaris*-specific IgE levels. Tuberculin induration was found to be inversely related to IgE in patients but not in controls.
[65]	Human study in West Cameroon. *Onchocerca volvulus*	To determine total serum IgE before and after tuberculosis therapy	T.B. therapy resulted in reduced serum *Ascaris*-specific IgE levels. Tuberculin induration was found to be inversely related to IgE in patients but not in controls.
[64]	Human study in East Ethiopia. *Ascaris lumbricoides, hookworms, Trichuris trichiura, Strongyloides stercoralis, Hymenolepis nana* and *Taenia* spp.	To investigate the effect of intestinal helminths on the immune response to PPD in naturally immunized or BCG-vaccinated individuals	Individuals who received BCG vaccination and were infected with helminths had reduced T cell and PPD skin test responses. Increased T cell proliferation and IFN were associated with improved BCG efficacy following anthelmintic therapy.
[66]	An experimental study in Ethiopia. *Schistosoma mansoni*	To investigate whether chronic helminth-infected individuals have reduced efficacy of BCG vaccine compared to uninfected persons	Possibly through attenuation of protective immune responses to mycobacterial antigens and/or by polarizing the general immune responses to the Th2 profile, *S. mansoni* infection reduced the protective efficacy of BCG vaccination against *Mtb*.
[53]	Human study in Ethiopia. *Ascaris lumbricoides, Hookworm, Strongloides stercoralis, Trichuris trichiura, S. mansoni* and *Enterobius vermicularis*	To study the prevalence of intestinal helminth infections and their association with active T.B. in T.B. patients and healthy household contacts	In addition to HIV infection, intestinal helminth infection may be a risk factor for the development of active pulmonary T.B. This discovery could have significant consequences for the control of tuberculosis in helminth-endemic areas around the world.
[30]	Human study in Ethiopia. *Trichuris trichiura, Ascaris lumbricoides, hookworms, Taenia* spp., *Hymenolepis nana* and *Enterobius vermicularis*	This study tested anti-helminthic medication before BCG vaccination to determine if it could improve BCG vaccination immunogenicity in helminth-infected patients	Chronic worm infection reduced BCG immunogenicity in humans. This was linked to increased TGF-β production but not a better Th2 immune response.
[74]	Human study in South Africa. *Ascaris lumbricoides* and *Trichuris trichiura*	To investigate whether helminth infection could affect a child’s ability to generate a proper Th1 immune response, which was defined by a positive tuberculin skin test (TST)	Helminth infection/exposure may reduce the immune response to *Mtb* infection. In younger children, being Ascaris IgE-positive significantly reduced the likelihood of being TST-positive, but this effect faded as they grew older.
[75]	Human study in Venezuela. *Ascaris lumbricoides* and *Trichuris trichiura*	To investigate the effects of parasite infections, malnutrition and plasma cytokine profiles on tuberculin skin test (TST) positivity	TST positivity was associated with low plasma Th1 cytokine levels in indigenous Venezuelan children with T.B. contacts and helminth infections.
[19]	Animal study in South Africa. *Nippostrongylus brasiliensis (Nb)*	To investigate the impact of acute *Nb*-induced lung damage and long-term parasite lung conditioning on the host’s ability to control mycobacterial infection	The findings show that early stage *Nb* infection induces a macrophage response that protects against subsequent mycobacterial infection.
[76]	Human study in Ethiopia. *Giardia lamblia, Ascaris lumbricoides, Hookworm* spp., *Strongyloides stercoralis, Trichuris trichuria, Enterobius vermicularis, Taenia* spp., *Hymenolepis nana, Schistosoma mansoni* or trophozoite stage of *Entamoeba histolytica.*	To diagnose latent *Mtb* infection using the tuberculin skin test (TST) and the IFN-γ release assays in helminth infected school children	The tuberculin skin test should be used with caution in areas where parasitic intestinal infections are common.
[77]	Human study in Uganda. *Hookworm, Trichuris trichiura, Hymenolepis nana, Schistosoma mansoni, Ascaris lumbricoides, Hymenolepis nana* and *Schistosoma mansoni*	To determine whether coinfections such as helminths, malaria and HIV modulate the immune system and increase susceptibility to latent tuberculosis infection (LTBI), leading to the persistence of the tuberculosis epidemic	Concurrent helminth, malaria and HIV infections did not affect cytokine responses profile in individuals with LTBI.
[78]	Human study in Ethiopia. *Schistosoma mansoni*	To investigate whether maternal helminth infection affects maternal and neonatal immunological function and T.B. immunity	The combination of early secretory antigenic target 6 (ESAT-6) and culture filtrate protein 10 (CFP-10) elicited a significantly lower IFN-γ response in helminth-positive than in helminth-negative participants. Cord blood mononuclear cells’ (CBMCs) IFN-γ response, total IgE and cross-placental transfer of T.B.-specific IgG were all negatively correlated with maternal helminth infection.
[17]	Human study in Ethiopia. *Ascaris lumbricoides Hookworm* spp. *Strongyloides stercoralis Trichuris trichiura* *Hymenolepis nana* *Taenia* spp.	To examine the clinical and immunological effects of helminth infection on T.B.	Asymptomatic helminth infection had a profound influence on the immunological profile of individuals with T.B. This favored Th2 immune responses such as increased regulatory T cells and IL-5 and IL-10 secreting cells.
[79]	Human study in Ethiopia. *Ascaris lumbricoides*	To investigate the clinical and immunological outcomes of patients coinfected with helminths and T.B. after albendazole treatment	The decrease in eosinophil counts and IL-10 demonstrated that asymptomatic helminth infection considerably impacts host immunity during tuberculosis and can be efficiently reversed with albendazole treatment. Helminth infection has clinical effects on chronic infectious diseases such as tuberculosis, and these effects should be further explored.
[80]	An animal study in the USA. *Schistosoma mansoni*	To investigate whether *Mtb*-specific T cell responses can be reversibly impaired by treatment of *S. mansoni* coinfection, without impacting arginase-1-expressing macrophage-mediated T.B. control	Anthelminthic treatment improved *Mtb*-specific T cell responses. In T.B.-infected mice, arginase-1-expressing macrophages in the lung formed granulomas and exacerbated inflammation.
[81]	An experimental animal study in USA. *Heligmosomoides polygyrus*	To investigate whether *Mtb* infection would be modulated in mice with chronic *H. polygyrus* infection	Despite a systemic increase in FoxP3+ T regulatory cells, neither primary nor memory immunity conferred by Mycobacterium bovis BCG vaccination were affected in mice with chronic enteric helminth infection.
[82]	Human study in India. *Strongyloides stercoralis*	To investigate whether helminth modulation of cytokine responses in latent T.B. coinfection is reversible after anthelminthic therapy	In *Strongyloides stercoralis*-latent T.B. coinfection, anthelmintic therapy reversed the modulation of systematic and T.B. antigen-stimulated cytokine responses.

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
