# Peer review of "Immunological Interactions between Intestinal Helminth Infections and Tuberculosis"

_diagnostics, 2022, doi:10.3390/diagnostics12112676_

Round 1

Reviewer 1 Report

This article has an attractive title, but it seems that more references should be used:

1. Are intestinal helminths playing a positive role in tuberculosis risk? A systematic review and meta-analysis

2. Immunocompromised patients with pulmonary tuberculosis; a susceptible group to intestinal parasites

3. Frequency, associated factors and clinical symptoms of intestinal parasites among tuberculosis and non-tuberculosis groups in Iran: a comparative cross-sectional study

4. Blastocystis subtype 1 (allele 4); predominant subtype among tuberculosis patients in Iran

5. Toxoplasma gondii seroprevalence among tuberculosis patients: A systematic review and meta-analysis

Author Response

REBUTTAL:

                                                              Manuscript Number: Diagnostics 1943338

Dear Reviewer

Thank you for the constructive comments. All the revised work on the manuscript has been highlighted in yellow.

Reviewer #1:

Comments from the reviewer:  This article has an attractive title, but it seems that more references should be used

  1. Are intestinal helminths playing a positive role in tuberculosis risk? A systematic review and meta-analysis.

The above article was added. Please refer to lines 393 - 395.

  1. Immunocompromised patients with pulmonary tuberculosis; a susceptible group to intestinal parasites.

The above article was added. Please refer to lines 395 - 397.

  1. Frequency, associated factors and clinical symptoms of intestinal parasites among tuberculosis and non-tuberculosis groups in Iran: a comparative cross-sectional study.

The above article was added. Please refer to lines 406 – 408.

  1. Blastocystis subtype 1 (allele 4); predominant subtype among tuberculosis patients in

The above article was added. Please refer to lines 397 - 400.

  1. Toxoplasma gondii seroprevalence among tuberculosis patients: A systematic review and meta analysis

The above article was added. Please refer to lines 409 - 411.

Reviewer 2 Report

In the manuscript entitled “immunological interactions between intestinal helminth infections and tuberculosis” the authors present a scoping review on an important and under-researched theme, which I think will be of interest to a broad range of global health researchers. The authors write in good clear English; however, the content is, in my view, poorly organized, repetitive, and discursive. Presently the review is formatted as a systematic review with a methods, results and discussion section. As the manuscript is clearly not a systematic review (diagnostics require such reviews to follow PRISMA guidelines which this document does not), I suggest that the authors reformat the review so that it is in a more conventional scoping review format. In my view, a more conventional review format, using extensive sub-titling, could help the authors to give their review a clearer direction, help them to avoid repetition and organize their content.

Author Response

REBUTTAL:

                                                              Manuscript Number: Diagnostics 1943338

Dear Reviewer

Thank you for the constructive comments. All the revised work on the manuscript has been highlighted in yellow.

Reviewer # 2:

Comments from the reviewer:  In the manuscript entitled “immunological interactions between intestinal helminth infections and tuberculosis”, the authors present a scoping review on an important and under-researched theme, which I think will be of interest to a broad range of global health researchers. The authors write in good clear English; however, the content is, in my view, poorly organized, repetitive, and discursive. Presently the review is formatted as a systematic review with methods, results and a discussion section. As the manuscript is clearly not a systematic review (diagnostics require such reviews to follow PRISMA guidelines which this document does not), I suggest that the authors reformat the review so that it is in a more conventional scoping review format. In my view, a more conventional review format, using extensive sub-titling, could help the authors to give their review a clearer direction, help them to avoid repetition and organize their content.

The comments are noted – The format of the abstract was amended. Please refer to lines 24 – 39.

A PRISMA diagram was added. Please refer to lines 112 – 114.

Subtitling/ numbering of the sections on the manuscript was done.

Reviewer 3 Report

This paper provides an important review of available literature on the immunologic interactions between two highly endemic infectious diseases in developing countries, helminths and tuberculosis. The authors provided a good summary on the immunology of both single and concomitant infections. As we push for the elimination of both diseases, understanding the their interaction can improve our understanding of real world scenario, wherein people are not infected with a single pathogen. However, in its current form, the paper does not clearly discuss this interaction.

Here are some comments to improve the manuscript:

Abstract

Line 32: change summarizes to summarized

Line 49: change 1,5 billion to 1.5 billion

Line 58: Clarify if Schistosoma mansoni is included in this review since it is not an intestinal helminth as it resides in the mesenteric veins.

Lines 75-89: Combine into one paragraph

Method
Line 111: Specify which online databases were accessed and the time of publications.

Discussion
Line 115: Clarify if this is the start of discussion.

Line 117: Remove chronic and extracellular in the description. 

Line 126: Suggest to revise: Majority of helminths do not replicate within the host...

Line 137-143: Combine in one paragraph

Line 153: Change to "...and hyperplasia of goblet and mucosal mast cells."

Line 183: Change eradicated to eliminated

Line 201: Suggesting to change color combination of the figure to make it more reader friendly; clarify if arrows from TH1 to Treg is going inwards or should be outwards; suggesting to change from late to chronic infection

Line 320: Suggesting to change color combination of the figure to make it more reader friendly; might want to consider placing how latent state is converted to active TB disease

Line 364: Please state meaning of AAM during first mention and no need for succeeding mentions.

Line 439: Italicize S. mansoni. Please use abbreviation of genus when used more than once.

Line 464: Please clarify statement. Please combine into one paragraph.

Line 516: Change "by PBMC" to "of PBMC"

General comment:
1. Can you clarify if discussion on immunological interaction is based on assumption that individuals get helminthiasis first then TB?

2. Clarify if review will include Schistosoma mansoni and tissue helminths, since this is not an intestinal helminth. If included, please change topic to helminth in general (and not intestinal helminths).

3. Suggest to include some information, if available, about triple infection with STH, TB, and HIV. This will improve the impact of the paper.

4. It is important to emphasize about impact of con-infection to immune-mediated pathology, and not only to immunoprotection (vaccination).

Author Response

REBUTTAL:

                                                              Manuscript Number: Diagnostics 1943338

Dear Reviewer

Thank you for the constructive comments. All the revised work on the manuscript has been highlighted in yellow.

Reviewer # 3:

Comments from the reviewer:  This paper provides an important review of available literature on the immunologic interactions between two highly endemic infectious diseases in developing countries. The authors provide a good summary on the immunology of both single and concomitant infections. As we push for the elimination of both diseases, understanding their interaction can improve our understanding their interaction can improve our understanding of real-world scenario, wherein people are not infected with a single pathogen. However, in its current form the paper does not clearly discuss this interaction. Here are some comments to improve the manuscript.

Abstract

Line 32: change summarises to summarized

The change was made. Please refer to line 31

Line 49: change 1,5 billion to 1.5 billion

The change was made. Please refer to line 46.

Line 58: Clarify if Schistosoma mansoni is included in the review since it is not an intestinal helminth as it resides in the mesenteric veins.

Schistosoma mansoni is included in the review. The WHO fact sheet dated 8 January 2022, accessed on the 13th of October 2022, classifies Schistosoma mansoni as causing intestinal schistosomiasis (WHO, 2022).

Reference

WHO (2022) Schistosomiasis. Available at: https://www.who.int/news-room/fact-sheets/detail/schistosomiasis (Accessed: 13 October 2022).

Lines 75 – 89: Combine into one paragraph

The changes were made. Please refer to lines 71 – 82.

Method

Line 111: Specify which online databases were accessed and the time of publication.

The changes were made. Please refer to lines 105 -106 and Figure 1 – PRISMA flow diagram, lines 111-113.

Discussion

Line 115: Clarify if this is the start of discussion.

Please note that this section is the beginning of the explanation on the immunity to helminths. Please refer to line 116.

Line 117: Remove chronic and extracellular in the description.

The changes were made. Please refer to line 118.

Line 126: Suggest to revise: Majority of helminths do not replicate within the host…

The changes were made. Please refer to lines 127 – 129.

Line 137 – 143: Combine in one paragraph

The changes were made. Please refer to lines 138 -143.

Line 153: Change to “…hyperplasia of goblet and mucosal mast cells”.

The changes were made. Please refer to lines 153 - 154.

Line 183: Change eradicated to eliminated

The changes were made. Please refer to line 183

Line 201: Suggesting to change colour combination of the figure to make it more reader friendly; clarify if arrows form Th1 to Treg is going inwards or should be outwards; suggesting to change from late to chronic infection.

The changes were made. Please refer to lines 200 – 201.

A foot note was inserted to explain the arrows. Please refer to lines 218 -220.

Line 320: Suggesting to change colour of the figure to make it more reader friendly; might want to consider placing how latent is converted to active T.B. disease.

The changes were made. Please refer to lines 322 - 323

Line 364: Please state the meaning of AAM during first mention and no need for succeeding mentions.

The changes were made. Please refer to lines 369, 372 and 376.

Line 439: Italicize S.mansoni. Please use the abbreviation of the genus when used more than once.

The changes were made. Please refer to lines 403, 420 and 422.

Line 464: Please clarify the statement. Please combine them into one paragraph.

The changes were made. Please refer to lines 480 – 481.

Line 516: Change “by PBMC to of PBMC”

The change was made. Please refer to line 532.

General comment

  1. Can you clarify if discussion on immunogical interaction is based on assumption that individuals get helminthiasis first then TB?

The immunological interactions are bidirectional and are unaffected by whether helminths or T.B. are acquired first.

Clarify if review will include S. mansoni and tissue helminths since this is not an intestinal helminth. If included please change topic to helminth in general (and not intestinal helminths).

Schistosoma mansoni is included in the review. The World Health Organasation (WHO) fact sheet dated 8 January 2022, accessed on the 13th of October 2022, classifies Schistosoma mansoni as causing intestinal schistosomiasis (WHO, 2022).

Reference

WHO (2022) Schistosomiasis. Available at: https://www.who.int/news-room/fact-sheets/detail/schistosomiasis (Accessed: 13 October 2022).

  1. Suggest to include some information, if available, about triple infection with STH, TB and HIV. This will improve the impact of the paper.

The changes were made. Please refer to lines 470 – 473.

  1. It is important to emphasize about impact of co-infection to immune mediated pathology, and not only on immunoprotection (vaccination).

The changes were made. Please refer to lines 531 - 539

Round 2

Reviewer 3 Report

The manuscript have improved with the revisions of the authors. 

Please edit the figures to improve readability.